# Persistent multi-scale fluctuations shift European hydroclimate to its millennial boundaries

Y. Markonis [1], M. Hanel [1], P. Máca[1], J. Kyselý[1,2] & E.R. Cook[3]

In recent years, there has been growing concern about the effect of global warming on water resources, especially at regional and continental scales. The last IPCC report on extremes states that there is medium confidence about an increase on European drought frequency during twentieth century. Here we use the Old World Drought Atlas palaeoclimatic reconstruction to show that when Europe's hydroclimate is examined under a millennial, multi-scale perspective, a significant decrease in dryness can be observed since 1920 over most of central and northern Europe. On the contrary, in the south, drying conditions have prevailed, creating an intense north-to-south dipole. In both cases, hydroclimatic conditions have shifted to, and in some regions exceeded, their millennial boundaries, remaining at these extreme levels for the longest period of the 1000-year-long record.

[1] Faculty of Environmental Sciences, Czech University of Life Sciences Prague, Kamýcká 129, Praha - Suchdol 165 00, Czech Republic. [2] Institute of Atmospheric Physics, Czech Academy of Sciences, Prague 141 31, Czech Republic. [3] Lamont-Doherty Earth Observatory, Palisades, NY 10964, USA. Correspondence and requests for materials should be addressed to Y.M. (email: markonis@fzp.czu.cz)

Understanding the spatial and temporal patterns of the hydroclimatic variability has been one of the most challenging subjects in contemporary climatic and hydrological research due to its numerous components and the non-linear processes involved[1-4]. Physically based models cannot adequately represent the hydroclimatic system complexity[5,6], especially when it comes to regional patterns[7,8] and their links to atmospheric circulation[9]. Therefore, it is not surprising that there is an ongoing discussion about whether the frequency and severity of droughts is globally increasing[1,10,11] or whether the rise in global temperature has resulted in an increase in dryness over the arid regions and increase in wetness over the humid ones[12-15]. Palaeoclimatic reconstructions could potentially serve as the required basis not only for model-observation integration, but also for probabilistic inference, e.g., for estimation of return period of extreme events. A prominent case is the Old World Drought Atlas (OWDA[16]), a tree-ring reconstruction of the self-calibrated Palmer Drought Severity Index (scPDSI) over Europe and parts of Northern Africa and Middle East for the warm season of the last 2000 years (June to August). The OWDA follows the methodology applied in similar studies about hydroclimatic conditions of North America[17] or Asia[18] and presents some multidecadal periods of drought and wetness over large areas of Europe[16-18].

In this study, we scrutinize empirical evidence from the OWDA to attain more insight into current hydroclimatic regime of Europe and specifically placing it in its long-term context. Interestingly, Europe has been one of the regions with the most contradicting reports concerning recent changes in its hydroclimatic conditions and their extremes, i.e., droughts and floods[1]. There is an overall agreement that northern latitudes are getting wetter, while the south is moving towards the opposite direction[5,10,11,14,19,20]. However, the limited length of observational records hinders our ability to assess the significance of the observed change in the long-term context. The OWDA, or similar reconstructions, could help us determine its significance, by comparing the recent variability to the multidecadal fluctuations of the past.

## Results

**Multi-scale fluctuations and current conditions.** Since the beginning of the millennium, European hydroclimate had been fluctuating between pluvial and dry conditions (Fig. 1a; solid line). Several departures from the average conditions can be identified at the climatic scale (30 years), according to their severity and spatial extent (median scPDSI below 0.3 and ratio of grid cells above 0.7; see Methods). Amongst them, four correspond to wet and three to dry conditions, revealing two distinct features; the dry intervals generally last longer than their wet counterparts, and during the last 90 years wet conditions have prevailed, exhibiting more persistent behaviour than in the past. The only wet interval with similar duration of the current pluvial can be found in 1113–1172, but the 30-year deviations were not so severe and spatially spread.

When examined at the regional level, these periods seldom begin and end simultaneously or have the same severity at each individual region (Supplementary Fig. 1). This heterogeneous propagation is confirmed in the observational records and similar studies on European droughts[1,19,21]. The recent increase in wetness is observed mainly in the British Isles, Scandinavia, France and Central and Eastern Europe. Even though there was also a likewise rise in the Iberian Peninsula and Mediterranean Sea during the 1953–1982 interval, it was succeeded by three rather dry decades, which may be unprecedented in severity over the eastern Mediterranean[22] and possibly linked to global

warming due to increased evapotranspiration[23]. These findings are in good accordance with the changes of precipitation in CRU, GHCN and GPCC datasets, as presented in the last IPCC WGI (Intergovernmental Panel on Climate Change Working Group I) Report[5].

The non-exceedance probability of current 30-year pluvial conditions is found to be unique in terms of duration for the last 1000 years over most of Europe (Fig. 1a). This is not necessarily linked with events in finer time scales. For instance, during 1953–1982, the percentage of annual values with non-exceedance probability below 0.05 is quite close to its mean value, while the 30-year scPDSI is almost equal to the one of the 1983–2012 period. This is a common feature with the 1263–1292 and 1563–1592 pluvials. On the contrary, 1113–1142 pluvial had elevated annual maxima (above 0.1), although the 30-year conditions were not so extreme in terms of mean of the non-exceedance probability (Fig. 1a). Thus, an extreme 30-year period is not necessarily constituted of subsequent extremes in finer scales or vice versa.

Therefore, the role of time scale in the occurrence of extreme events should be considered. For instance, there are periods when extremes propagate in different time scales (1100–1200 for drought or 1550–1700 for pluvial conditions; Fig. 1b), suggesting scale-dependent fluctuations and/or periods of hydroclimatic polarization. The latter could occur either in the time or the space domain, i.e., simultaneous droughts and pluvials in different regions of Europe or shifting between dry and wet periods in finer temporal scales (e.g., 5 years of annual maxima followed by 5 years of minima, resulting in an aggregated 10-year period close to the mean). Such behaviour also manifests in the multidecadal progression of the current pluvial (Fig. 2; blue arrow). In addition to the steady increase in wetness, substantial polarization of the hydroclimatic conditions can be observed during the 1983–2012 interval, seen as a second peak close to 0 in Fig. 2 (see also Supplementary Fig. 2). Further investigation of the dynamics of the hydroclimatic conditions in the spatial domain showed that the second peak is due to the strong increase of dryness over the Iberia, the Balkans and some parts of central Europe (Fig. 3; see also Supplementary Fig. 1b).

In terms of severity, the current conditions are not the most extreme encountered (Supplementary Fig. 3). What makes them unprecedented is their coherent and extensive areal coverage, which lasts for 90 years. The only other 30-year interval that clearly surpasses them is the one between 1713 and 1742, having a median grid cell scPDSI quantile equal to 0.81, which is higher than 0.76 corresponding to the 1983–2012 interval. The 1713–1472 interval is also the only pluvial comparable to the dry intervals in terms of spatial extent before 1920, as the latter affect larger fractions of the European continent (Fig. 2, high-lighted rectangle). These 30-year dry intervals can be found within the 1023–1082, 1413–1472 and 1803–1862 periods (Supplementary Fig. 1a).

Another noteworthy result stemming from the scale analysis of extremes considers the relationship between the recent wet period and the increase in global (and continental) temperature. The spatial pattern of anomalies (Fig. 3) in the early twentieth century is very similar to the current period and follows the global temperature change, i.e., increase during 1923–1952 and 1983–2012 and slight decline during 1953–1982. Notably, the areal extent of pluvial conditions fluctuates similarly in all but the 30-year time scale (Fig. 2b). This suggests that during this period there are, at least, two main drivers of hydroclimatic variability; one following directly the temperature dynamics affecting finer scales, and the other manifesting in the longer ones. This slower component seems to become progressively dominant and in 1983–2012 the cell percentage increases with scale for both

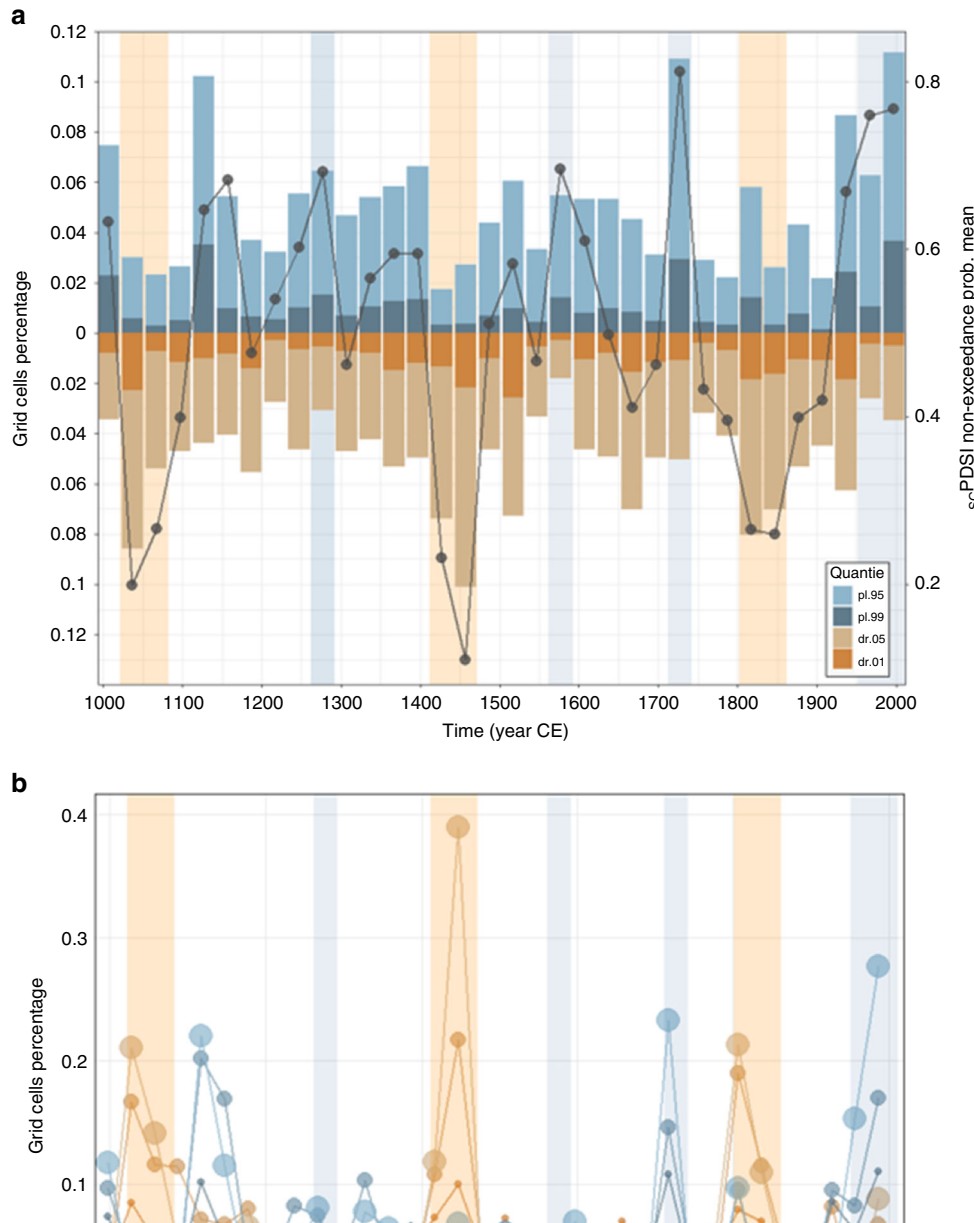

**Fig. 1** Long-term hydroclimatic change over Europe. **a** Mean of the non-exceedance probability (MNP) of the scPDSI for Europe at 30-year scale (black circled line) and percentage of grid cells with scPDSI in the top/bottom 0.05 and 0.01 quantiles per 30-year periods (blue and brown bars). Light orange and blue areas correspond to droughts and pluvials in the 30-year scale. **b** Percentage of grid cells with scPDSI in the top (bottom) 0.95 (0.05) quantile per 1-, 10- and 30-year period (blue and brown circles; size corresponds to scale, i.e., the smallest are in annual scale). Grey dashed line represents the nominal quantile value

pluvial and dry conditions. Similar findings have been reported in the study of drought in regional model simulations, where the future changes in the longer scales are found to be more pronounced than the shorter ones[24]. Another likewise example can be found in the well-studied California drought; the warming component of the drying process appears to be slowed down by natural variability, but still evident as a background trend[25].

On the other hand, during the 1923–1952 interval, the 30-year component is not manifested. Further investigation (Fig. 4) demonstrated that this is due to the succession of years with very high and very low number of grid cells with extreme values. This temporal polarization in the annual scale resulted in the complete lack of extremes in the 30-year scale. Since 1973, there are no years with no annual wet extremes, i.e., even in the driest years, some regions of Europe experience extreme wet conditions. This was found to be unprecedented in the whole record in terms of duration.

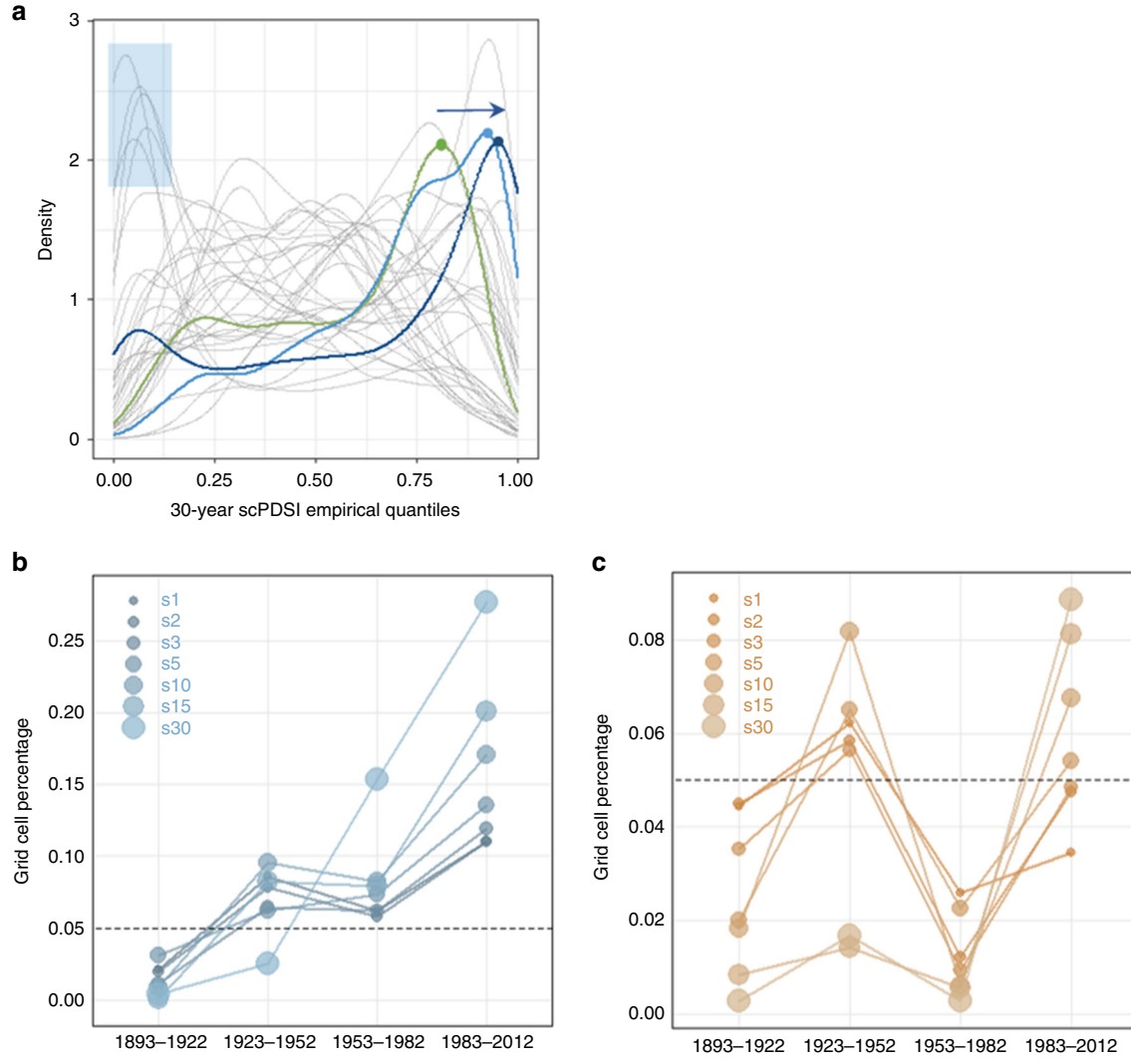

**Fig. 2** Temporal evolution of hydroclimatic conditions over 1892−2012. **a** Empirical probability densities of 30-year scPDSI values over Europe (grey lines: different 30-year intervals for the 992−1922 period, green line: 1923−1952 interval, light blue line: 1953−1982 interval, dark blue line: 1983−2012 interval). The blue arrow represents the temporal evolution of the last three 30-year intervals (green to light blue to deep blue). The dry periods are also highlighted by the blue transparent rectangle. Values close to zero (close to one) represent extremely dry (extremely wet) conditions. **b** Percentage of grid cells with scPDSI above the 0.95 quantile for time scales 1 to 30 years (extreme pluvial conditions). **c** Similar to **b** but for the bottom 0.05 quantile (extreme dry conditions). Grey dashed line represents the theoretical quantile value, i.e., the top/bottom 5% of the grid cells

Additionally, the comparison of the recent hydroclimatic temporal dynamics with the other pluvials advocates to the fact that the recent pluvial has the most persistent progression, and only during 1113–1142 and 1713–1742 the same pattern in the frequency of extremes across scale was observed (Fig. 5; see also https://shiny.fzp.czu.cz/KVHEM/OWDA/). Even though there were also positive temperature trends in these two periods, closer examination revealed other periods with similar thermal changes without a corresponding increase in the pluvial conditions. In any case, the aforementioned results highlight the need of analysing the influence of temperature on droughts/pluvials, especially regarding the future climatic change. Since the validity of current model simulations for precipitation is still significantly debated[26], we should try to understand to what extent the projected increase in temperature, which is currently better represented by Earth system models, might affect the forthcoming hydroclimatic regime.

**Links with temperature and NAO.** A straightforward way to assess the observed change could be to regard it as a consequence

of the global water cycle intensification due to global warming, commonly referred to as the "wet-get-wetter and dry-get-drier" paradigm[12]. Stemming from the direct application of Clausius-Clapeyron equation, an increase in temperature is expected to intensify the global water cycle due to the corresponding rise in atmospheric water holding capacity. This was demonstrated in the early modelling attempts of Earth's climate (e.g., the study of Washington and Meehl[27]) and is still supported by modern physically based models[28]. Recent observational studies, though, have doubted its validity, implying significant non-linear mechanisms and a more active role of atmospheric circulation especially over land[5,13–15].

The atmospheric circulation and specifically the North Atlantic Oscillation (NAO) could also be a possible explanation for the emerging substantial dipoles over Europe, such as the Iberian drying vs. Scandinavian wetting[29] (Fig. 3). However, closer examination of the spatial features of OWDA unfolds a rich compilation of multidecadal patterns that could hardly be assessed to specific modes of atmospheric circulation. In addition,

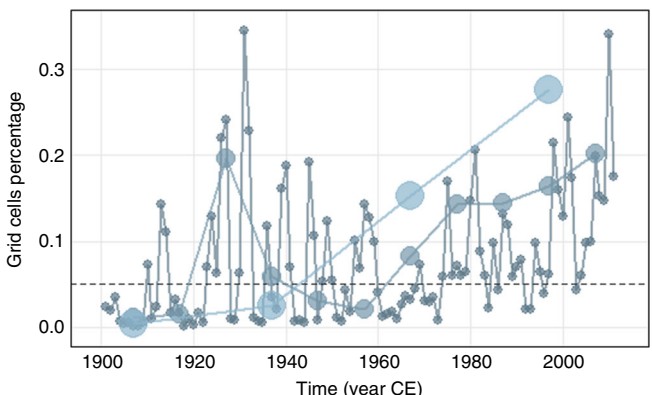

**Fig. 3** Spatial evolution of hydroclimatic conditions over 1892–2012. Empirical quantiles of 30-year scPDSI (**a** 1893-1922, **b** 1923-1952, **c** 1953-1982, **d** 1983-2012); values close to zero (close to one) represent extremely dry (extremely wet) conditions (AL Alps, BI British Isles, EA Eastern Europe, FR France, IP Iberian Peninsula, ME Mid Europe, MED Mediterranean, SC Scandinavia). The corresponding empirical distributions can be found in Supplementary Fig. 1a

**Fig. 4** Cross-scale examination of recent pluvial. Grid cell percentage above the 0.95 quantile at 1-, 10-, and 30-year scales since 1900. Grey dashed line represents the theoretical quantile value

while OWDA refers to warm season conditions, the NAO has a stronger and spatially more extended influence on the European climate during winter. Over summer, it is known to affect the western, mid-to-high latitude areas[30], and hence links to drought are less likely to be manifested at the continental scale.

Simple linear correlation-based methods are not sufficient for determination of the effect of temperature and atmospheric circulation to drought due to their restrictions by using a representative point-derived NAO index[31]. Hence, a logistic regression approach was applied and provided some insight for links between hydroclimatic conditions, regional temperature and decadal December−May NAO index between 1500 and 2012 (Fig. 6; Supplementary Fig. 4). The NAO and temperature effect appear to be coupled, although the former seems to have a more coherent pattern, mainly evident in western and northern Europe. At higher latitudes, decadal wet conditions appear to emerge more often when temperature is above average and the December−May NAO is positive. The reverse influence of summer NAO has also been identified in/over central and northern Europe (Supplementary Fig. 5), as presented in other palaeoclimatic

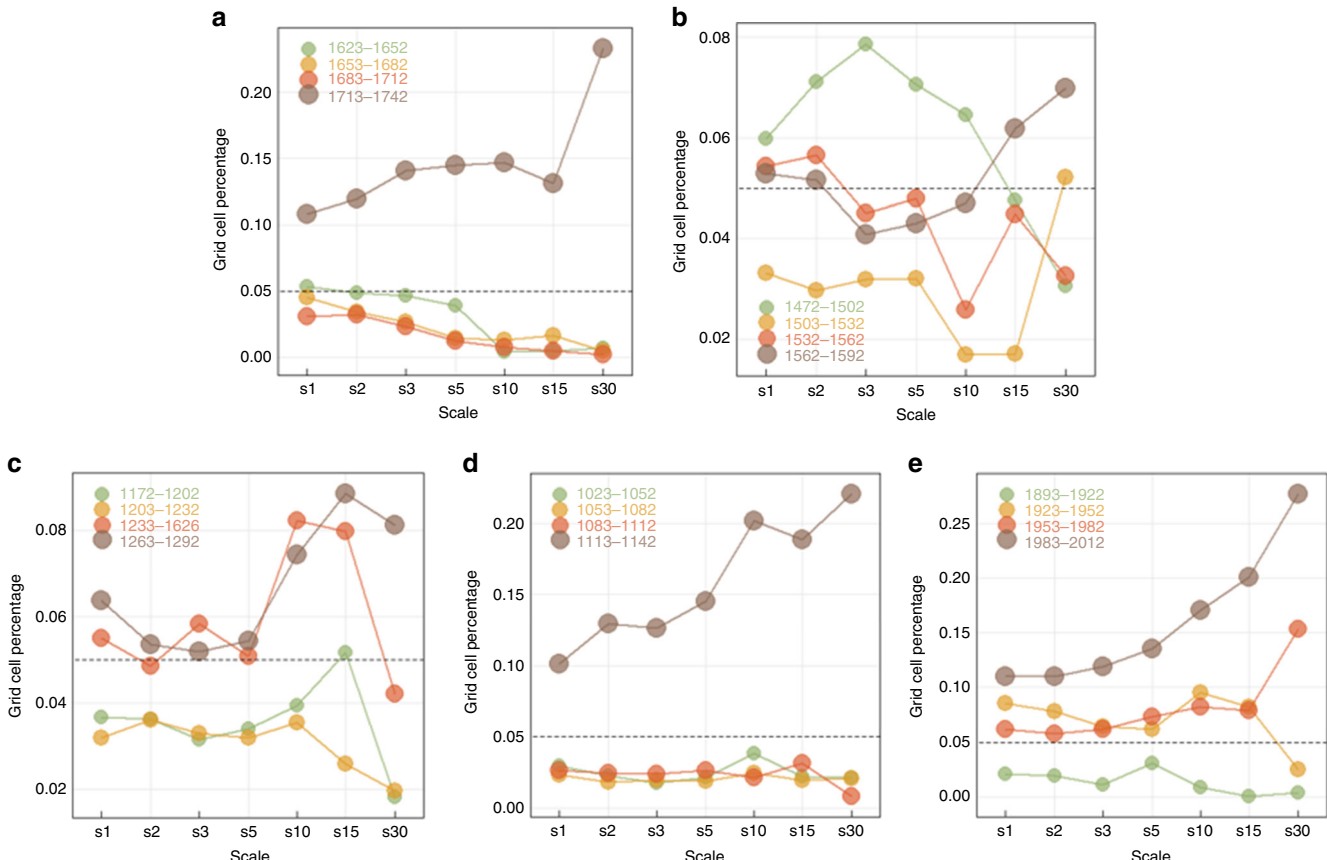

**Fig. 5** Cross-scale analysis of major pluvials. Temporal progression per 30-year interval in terms of grid cells percentage; s1 to s30 refer to different temporal scales, e.g., s1 is 1 year, s2 is 2 years, etc. Grey dashed line represents the theoretical quantile value. Each panel corresponds to a different period: **a** 1623-1742, **b** 1473-1592, **c** 1173-1292, **d** 1023-1142, **e** 1893-2012

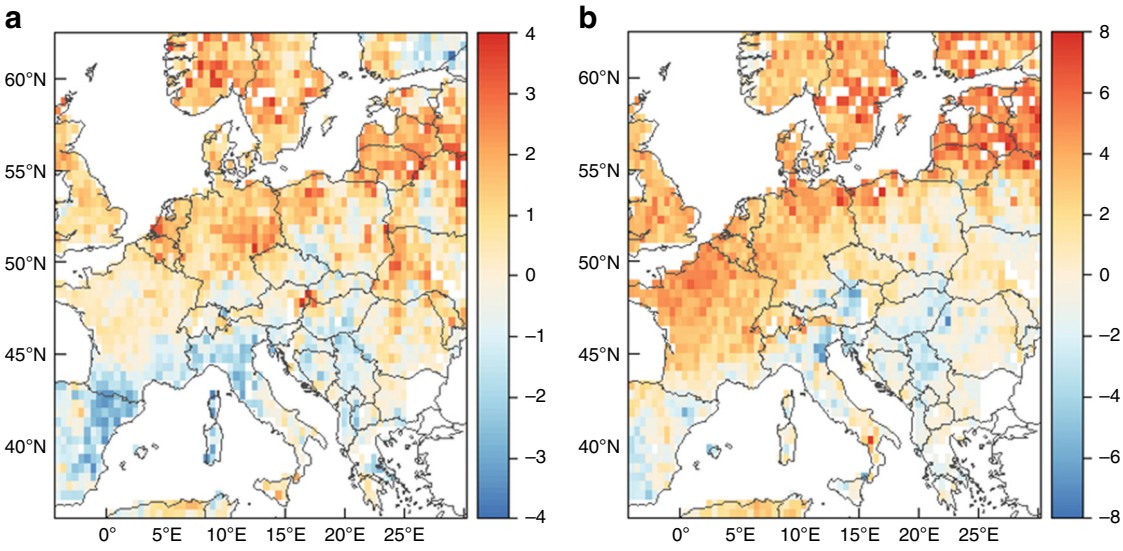

**Fig. 6** Logistic regression analysis. Non-linear relationships between 10-year scPDSI and **a** regional temperature and **b** seasonal NAO for 1500–2012, as presented by the logistic regression coefficients as estimated by Eq. 2 (see Methods; positive sign correspond to positive correlation). The corresponding statistical significance is presented in Supplementary Fig. 4

studies[32,33] and observational records[30]. In southern and eastern Europe, there is not any statistically significant signal for either temperature or NAO, with the exception of some regions of the Iberian Peninsula, Italy and the Balkans, where wet conditions are linked inversely with temperature. It should be pointed out that the removal of the last 90 years of data does not alter the link between wetness and temperature at northern latitudes (Supplementary Fig. 6).

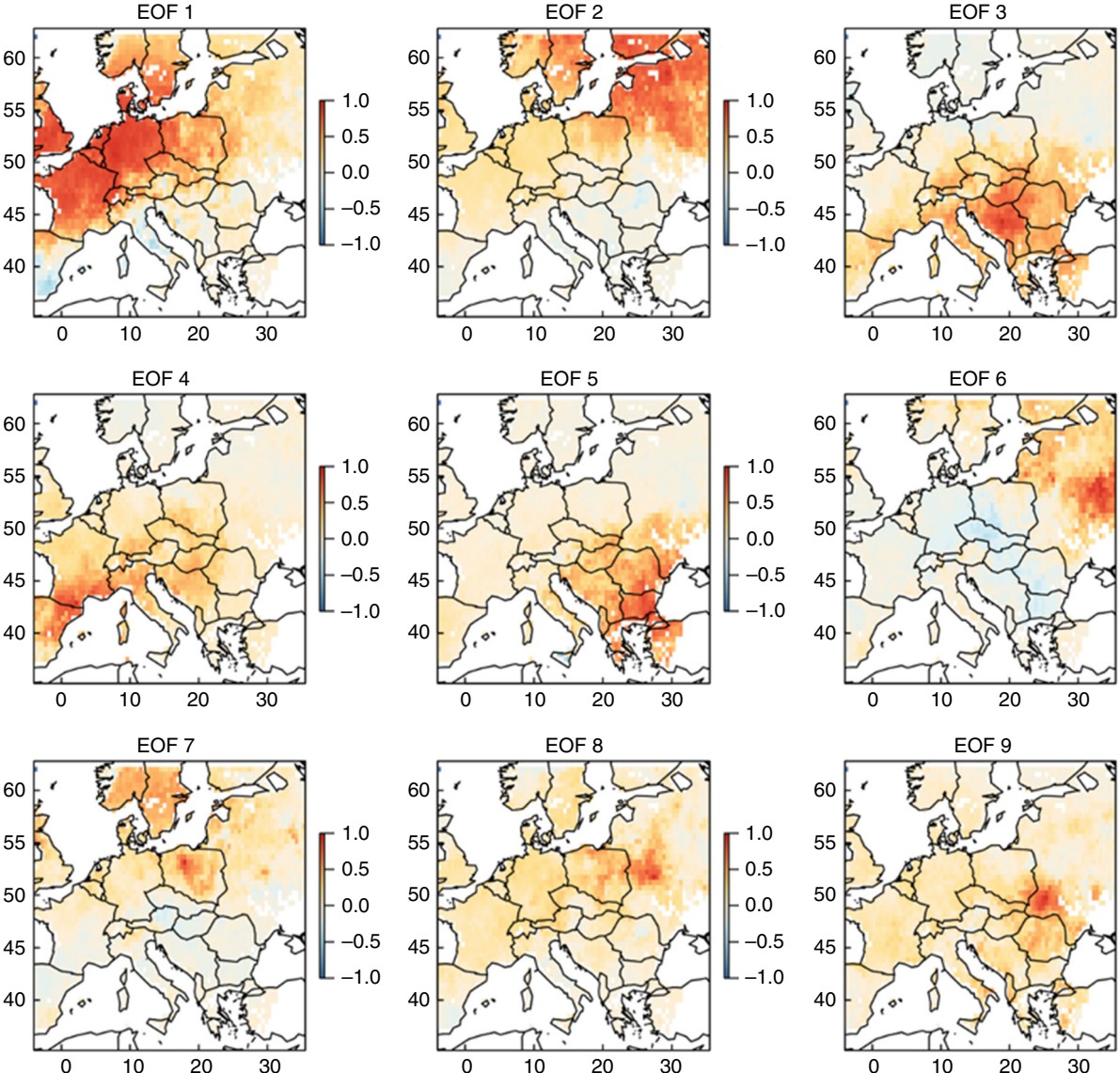

**Fig. 7** Empirical orthogonal functions. Spearman's linear correlation coefficients between scPDSI and the time series derived by empirical orthogonal functions

NAO is not the only circulation mode that is linked with European hydroclimate, although it is considered to be the most representative and physically relevant to Northern Hemisphere variability[34]. Other known modes include Arctic Oscillation (AO), the Scandinavian (SCA) and Eastern Atlantic/Western Russia (EA/WR) patterns[21,31,35–37]. However, since there is no palaeoclimatic reconstruction of these indices, it is difficult to robustly determine their influence on European hydroclimate through OWDA. To overcome this setback, one could compare the Empricial Orthogonal Function (EOF) patterns to the corresponding spatial correlation patterns of the observational records of the indices[38]. It should be acknowledged though that this is representative only for the periods that the global circulation retains the main features of the last 50 years, i.e., their observational record length. Encouragingly, recent findings claim that at least for two periods with shifted hydroclimatic conditions, the Medieval Climate Anomaly and the Little Ice Age, modern analogues of atmospheric circulation patterns are detected in Europe[39].

The first nine EOFs of the scPDSI have been determined[40], explaining approximately half of the observed variance (54%) and depicting some distinct spatial patterns (Fig. 7). It is worthy to note the striking analogy between EOF1/EOF2 and Fig. 6b, corresponding to the NAO correlation (33% of variance), while the EOF4 pattern can be found within the temperature regions of influence (Fig. 6a). On the contrary, the EOF modes above 6 represent rather local spatial patterns (6% of variance). The comparison between the AO and SCA correlation patterns with the CRU scPDSI dataset for the last 50 years implies that they share similar features with EOFs 4, 5 and 6 of OWDA scPDSI (8% of variance), confirming the previous studies[21,31,39]. Similar to the case of NAO, the distinction between the effects of circulation indices and temperature is quite obscure. For example, SCA (Supplementary Fig. 7a, c, e) seems to present the same pattern with temperature (Fig. 6a) and EOF4. On the other hand, most of the indices are correlated with temperature only for central to northern Europe (NAO is similar to AO; not shown here), pointing to different physical processes in hydroclimatic conditions over southern Europe. Correspondingly to the NAO, the rest of the major climatic indices confirm the strong interaction between temperature, precipitation and atmospheric circulation.

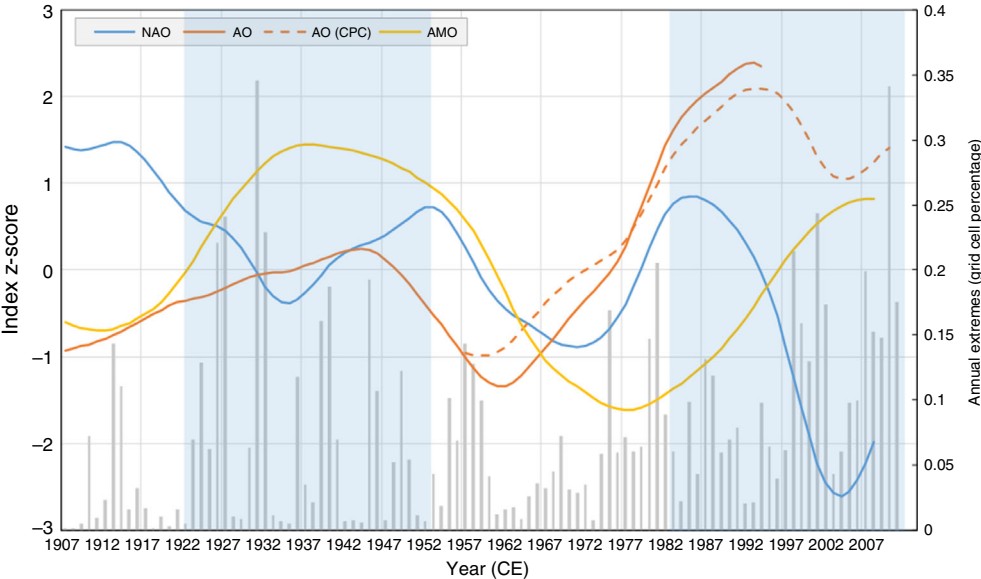

**Fig. 8** Annual pluvial extremes and major atmospheric/oceanic circulation indices. 30-year low-pass filter (loess) z-scores of annual North Atlantic Oscillation (NAO), Arctic Oscillation (AO, AO-CPC) and Atlantic Multidecadal Oscillation (AMO). All datasets extracted from the KNMI explorer online platform. Light blue shade represents the intervals 1923–1952 and 1983–2012. On the secondary vertical axis, the grid cell percentage of annual extremes (above the 0.95 quantile) is presented for comparison purposes (grey bars). The periods of spatial polarization are highlighted by transparent blue bars

## Discussion

Our deciphering of the OWDA suggests that during the last 100 years, the hydroclimatic conditions have persistently shifted. The observed change becomes more significant as the temporal and spatial scales get larger. The decadal regional averages present some extreme values since 1920, but when they are aggregated over a 30-year time window the picture changes dramatically. The last 30 years appear to be the wettest for the whole record in Scandinavia and the second wettest in France and Mid Europe, while they are the driest in the Iberian Peninsula (Supplementary Table 1). Similar high/low values emerge for other regions and intervals over the last 90 years, contributing to the persistence of the deviation from the millennial average and at the same time creating a polarized north-to-south pattern. Interestingly, this meridional divergence has also been identified in periods 1143–1172 and 1233–1262 (see also https://shiny.fzp.czu.cz/KVHEM/OWDA/), which lie within the warm period of Medieval Climate Anomaly. At that time, it is stated[41] that a persistent positive phase dominated NAO, which has also been observed during the 1900–1930 and 1970–1990 periods (Fig. 8).

The recent north–south polarization is also found to be reflected in the drought frequency and severity over southern Europe, driven primarily by the increase in temperature and the corresponding increase in evapotranspiration. Recent results depict a statistically significant trend towards less frequent and severe drought events over north-eastern Europe, mainly in winter and spring, while the opposite holds true for southern Europe, mainly in spring and summer[42,43]. Additionally, this pattern is in good agreement with drought projections of regional climate models for Europe[24,44]. Soil moisture availability has also been projected to further decline most in southern Europe in all seasons[45]. However, this does not necessarily imply that these hydroclimatic conditions will persist in the future.

The reason is that the physical processes involved in hydroclimatic change and the models used to describe it incorporate a thermal (or thermodynamic) and dynamic (or circulation) component. These two contributing factors are firmly interconnected. For instance, fluctuations in the atmospheric circulation could trigger changes in cloud cover and will then affect

surface radiative balance[46]. Likewise, a study of the EURO-CORDEX simulations indicates that the propagation of European summer heat waves could be linked to non-linear interactions between soil moisture and atmospheric circulation[47]. Nonetheless, when these components are decomposed[48], the north-to-south dipole is solely linked to the atmospheric circulation, whereas the thermal effect results in increased precipitation all over Europe, as well as integrated zonal transport of humid air masses from the Atlantic Ocean[49].

Hence, the persistence of the conditions that dominated European hydroclimate for the last 90 years strongly depends on the maintenance of the current circulation regime. However, the model projections on atmospheric circulation are quite ambiguous and we are still far from any confident statements about its evolution for reasons that are not expected to change anytime soon[50]. In two independent studies, it has been demonstrated that the north–south pattern of persisting wetness and dryness is reversed in individual ensemble members[51] or models[52], depending on the long-term evolution of atmospheric dynamics. To cope with this enhanced uncertainty, it has been proposed to explore different scenarios of changes in the large-scale circulation, namely the tropical and polar amplification of global warming, as well as the stratospheric vortex strength[52]. In the same study, the scenario that presents a large decline in the precipitation of southern Europe and synchronous increase in northern Europe involves the strengthening of the stratospheric vortex and a high tropical amplification of global warming. In contrast, other model results present increased precipitation over Mediterranean and reduced over central and northern Europe, implying that the precipitation volumes could return to their millennial means.

Our results confirm the strong coupling between the thermal and the dynamic effect. Since the OWDA is a reconstruction of self-calibrated Palmer drought index derived from tree-rings, inevitably the influence of soil moisture availability is also encompassed in the hydroclimatic signal of the proxy records. Therefore, it is hard to distinguish the explicit role of temperature vs. atmospheric circulation and establish links of causality between them. In light of the findings of Kröner et al.[48] on the

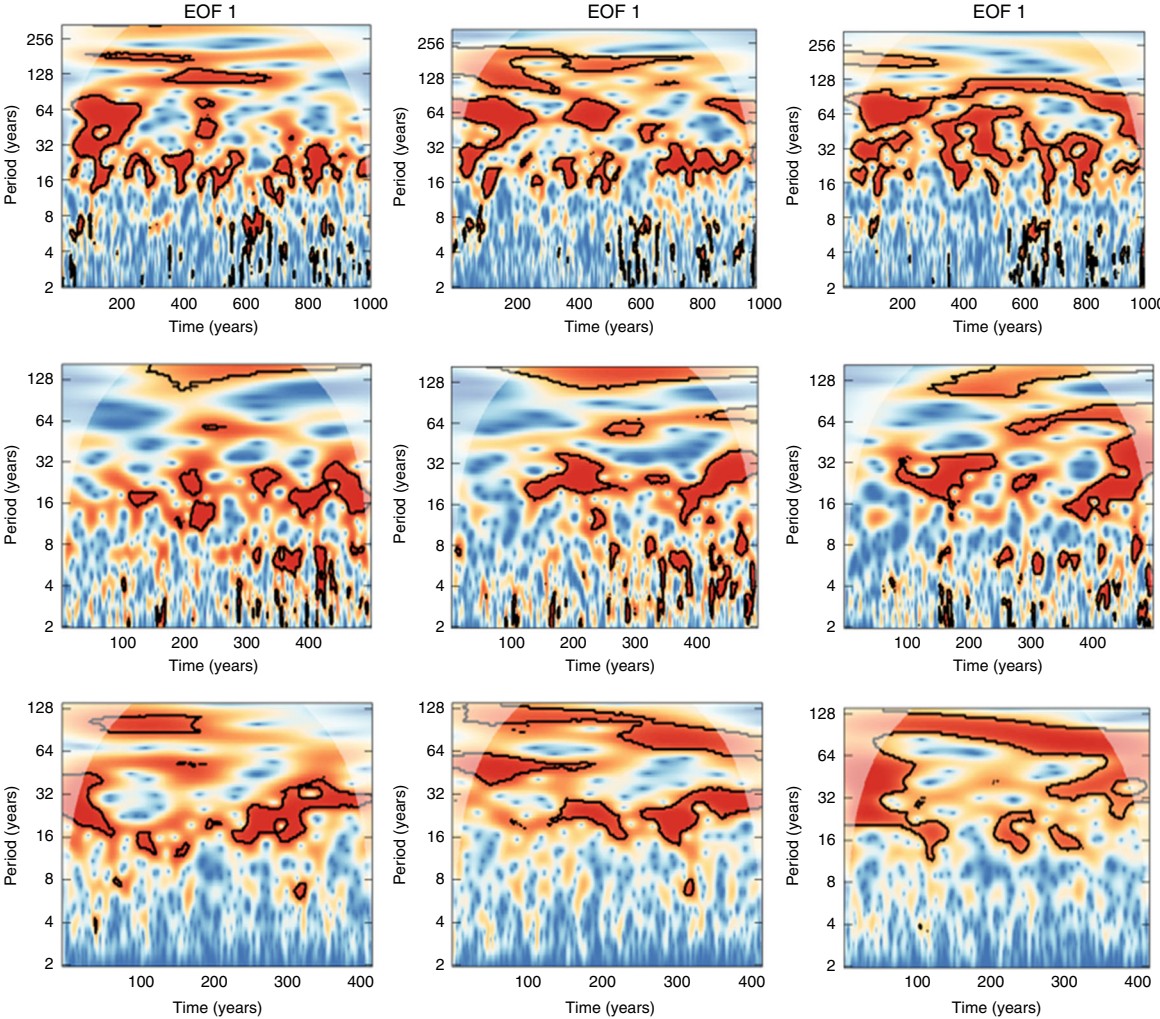

**Fig. 9** Cross-wavelet analysis between major hydroclimatic drivers and hydroclimatic patterns. Temperature (upper panel), winter NAO (mid panel) and AMO (lower panel) vs. the first three EOF modes of scPDSI. The black contour designates statistical significance ($p > 0.95$, assuming autoregressive noise spectra) and the cone of influence, where edge effects can potentially distort the picture, is shown as a lighter shade

decomposition of the thermal and circulation effect, though, we can highlight three different regimes. At high latitudes, the dominant wet conditions are driven by atmospheric circulation, amplified by the thermal effect in terms of both increased precipitation and zonal humidity transport. Here, these two components act synergistically (see also Fig. 6). Over France and Germany, there is a mixed impact of temperature and circulation, reflected also by conflicting results in relevant studies[48,51]. In southern Europe, the two components oppose each other, but the thermal effect does not compensate for the circulation effect. Additionally, there is a positive feedback of soil moisture to precipitation by reducing the regional recycling of water, which further amplifies the dry conditions[53].

In this description, the role of time scale should not be overlooked. Our exploration of the hydroclimatic fluctuations across the temporal continuum from years to multi-decades highlights that extremes in the lower frequencies are not always followed by extremes in the finer time scales. This is the case for the period 1953–1982, when the wet conditions prevailing at the 30-year scale are found to be extreme, whereas the annual values were quite lower (Fig. 4). On the contrary, the period 1923–1952 was characterized by numerous annual extremes, despite the fact that the 30-year average was close to its millennial mean. Therefore, there is a clear distinction between the subdecadal and

multidecadal fluctuations, which had been already identified for precipitation[54]. In the multidecadal domain, the centennial progression of current conditions is found to be unique in a millennial perspective (Fig. 5), implying that the observed hydroclimatic change could be the outcome of a fast and slow component.

The most plausible candidate for the slower component is the oceanic variability[55]. Consequently, the influence of the Atlantic oceanic circulation should also be considered. The Atlantic Multidecadal Oscillation (AMO) is known to influence summer European hydroclimate[56] or continental drought events in general[57]. Similar findings support the long-term correlation of AMO and drought in the United States[58], which has also been increasing during the second half of the twentieth century[59]. In addition, during both 1923−1952 and 1983–2012 periods, when the north−south dipole was evident, the AMO was shifting from negative to positive phase, whereas the opposite transition dominated the intermediate interval (Fig. 8). We have to note though that cryosphere, as well as vegetation changes, could also contribute significantly to the slower component and further research is needed to clarify this.

To further look into the interdependencies between time scales and the drivers of hydroclimatic variability, cross-wavelet analysis was applied and the cross-wavelet transformations of the three

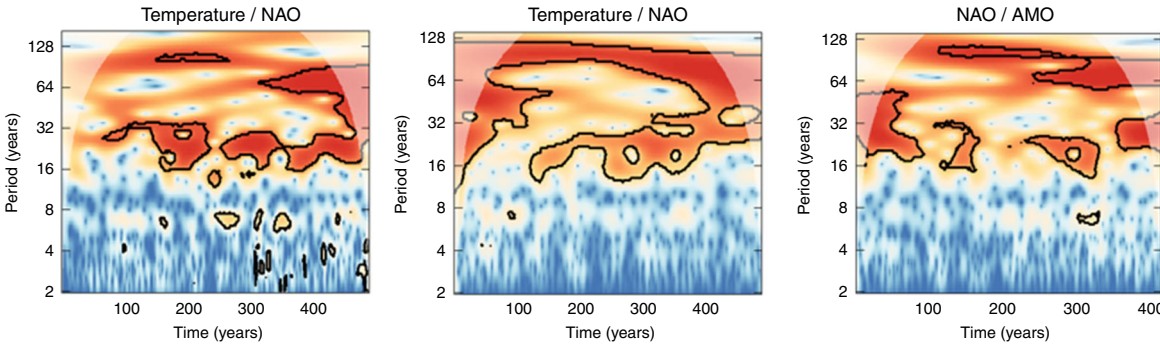

**Fig. 10** Cross-wavelet analysis between major hydroclimatic drivers. Similar to Fig. 9 but for cross-wavelets between temperature, NAO and AMO

most significant EOFs vs. temperature, NAO and AMO were explored (Fig. 9). It is easy to notice the difference between subdecadal and multidecadal behaviour in all three components and the strong covariance at the time scales between 16 and 32 years, which agrees with the logistic regression analysis. There is also limited covariance between NAO and the scPDSI at the 2- to 8-year domain, whereas there is no matching at scales between 32 and 64 years, where temperature and AMO are active.

When the wavelets of temperature, NAO and AMO are directly compared, their coupling at the time scales between 16–32 years and 64–128 years becomes even more apparent (Fig. 10). These findings justify even further our hypothesis that the atmospheric circulation effect is the main driver of Europe's hydroclimate, since AMO has been found to influence the surface air temperature of Northern Hemisphere[60,61] and the main driver of oceanic circulation was reported to be a combination of stochastic forcing from the mid-latitude atmospheric circulation[62] and aerosol forcing[63]. It has also been demonstrated that blocking activity and not NAO might be the best descriptor for climatic changes in the Atlantic Ocean[64], which might explain why NAO and AMO wavelets are not so strongly matched.

To conclude, we present evidence from observational and palaeoclimatic data showing that in millennial perspective, European hydroclimate is at the marginal band of its historical fluctuation (Figs. 1 and 2a). The extremity of the current multidecadal conditions involves a discrete north-to-south dipole, which follows both the twentieth century global temperature increase and the long-term atmospheric/oceanic circulation variability (Figs. 3 and 8). Most importantly, the steady, rather slow, progression of the phenomenon is unique when compared to the past pluvials (Fig. 5). Even though a complete description of the low-frequency drivers of European hydroclimatic variability remains elusive, the study of OWDA presents some evidence that atmospheric circulation could be the main driver of droughts/pluvials over Europe. In any case, we confirm that the "wet-gets-wetter and dry-gets-drier" paradigm[12] should be revisited, as claimed by Greve et al.[14] and explained by Byrne and O'Gorman[15], due to the complex interactions between temperature, atmospheric/oceanic circulation and precipitation. We have also shown how difficult it is to explicitly track down and quantify the effect of each individual component. Since there is general agreement in the scientific literature that the uncertainty in the projections of large-scale circulation and precipitation changes still remains high, it could be beneficial to move towards a more probabilistic-centred framework.

Finally, the implications of the presented multidecadal fluctuations are substantial for long-term water management and policy planning. It should be emphasized that the true range of hydroclimatic variability might remain hidden if only instrumental records are used, which is probably why many studies on temporal changes in drought yield spatially varied and often statistically insignificant results[5,10,11,14,19,20]. However, here we take advantage of the wider time window of the OWDA, which enhances information about the fluctuations of the recent hydroclimatic variability across different time scales. In such perspective, the recent shift in hydroclimatic conditions over Europe appears to be unprecedented over the last 1000 years.

## Methods

**Data.** The datasets included in this study in addition to the OWDA (https://www.ncdc.noaa.gov/paleo-search/study/19419) are the CRU v3.21 scPDSI gridded dataset of instrumental observations (1901–2012; https://climexp.knmi.nl/ CRU-Data/scPDSI.cru.3.24.bams.2016.GLOBAL.1901.2015.nc)[65], the reconstruction of European summer temperature gridded dataset (755–2011; https://www.ncdc.noaa.gov/paleo-search/study/19600)[66], the North Atlantic Oscillation (NAO) seasonal reconstruction (1500–1995; ftp://ftp.ncdc.noaa.gov/pub/data/paleo/pages2k/EuroMed2k/BHM_Eu2k2_Recon.nc)[67] and the reconstruction of Atlantic Multidecadal Oscillation (1567–1990; ftp://ftp.ncdc.noaa.gov/pub/data/paleo/treering/reconstructions/amo-gray2004.txt)[68]. OWDA has been compiled by the spatial point-by-point regression of 106 tree-ring chronologies to a map with 5414 half-degree grid cells[17]. In order to optimize the temporal and spatial distribution of our dataset, we have subsetted it according to two straightforward criteria: (a) the grid cell reconstructions should be based on at least 20 tree-ring chronologies within a 1000 km radius, as indicated by Cook et al.[17] and (b) all reconstructions should be based on tree-ring chronologies of similar length. The first rule was chosen to reduce the regression bias, while the second increased the sample homogeneity and allowed investigation of the spatial properties of hydroclimatic variability. The resulting data grid covered 35.25°N–62.75°N and 4.25°W–36.25°E (1940 grid cells; Supplementary Fig. 8) for the period 992–2012 and it was further spatially partitioned into the PRUDENCE regions[69] to inspect regional changes. In general, there was a good geographical match between the OWDA and the PRUDENCE regions (presented in Fig. 3), leaving out from the final data grid merely the western parts of the British Isles and the Iberian Peninsula.

To explore multi-scale fluctuations, the OWDA was temporally aggregated (averaged) to 10- and 30-year time steps, with corresponding sample sizes of 102 and 34 values, respectively, over the time period investigated. We have chosen the aggregating approach over other filtering methods (e.g., moving average or loess regression) to avoid altering the autocorrelation structure of the resulting time series and the potential introduction of significant bias to the quantile estimation. The determination of 30-year periods with significant deviations from the mean was performed on a 30-year scale basis for the whole Europe in terms of spatial median (due to skewness) and extent. Periods with median scPDSI below 0.3 (mean minus standard deviation rounded down) and spatial extent ratio over 0.7 (mean plus standard deviation rounded down) were classified as dry, while a median above 0.7 and a similar extent suggests wet conditions.

**Logistic regression analysis.** Non-linear links with multidecadal temperature and NAO were investigated by means of logistic regression, as described by Gareth et al.[70]. The latter was selected because simple linear techniques, e.g., Spearman's rank correlation, were found insufficient to describe potential relationships due to low signal-to-noise ratio. Logistic regression has been used extensively in ecological modelling for classification and modelling with quite robust results (see Hosmer et al.[71] and references therein).

To use logistic regression, all values of 10-year scPDSI above/below the 0.7/0.3 threshold were classified as dry/wet (set to 0 and 1 respectively), and then the probability of wet conditions $p(W)$ in each grid cell and each PRUDENCE region

was estimated through the logistic function:

$$p(W) = \frac{e^{\beta_0 + \beta_1 W}}{1 + e^{\beta_0 + \beta_1 W}} \qquad (1)$$

Here, $\beta_0$ and $\beta_1$ are the regression coefficients estimated by maximizing the likelihood function:

$$l\left(\hat{\beta}_0, \hat{\beta}_1\right) = \prod_{i:y_i=1} p(w_i) \prod_{i:y_i=0} (1 - p(w_i)) \qquad (2)$$

so that the estimated probability $\hat{p}(w_i)$ of wet conditions for each individual value of the descriptor variable $y_i$ is the most likely to describe the true one (Supplementary Fig. 9). For $y_i$, the 10-year regional temperature and NAO were selected. Regional temperature was averaged in each PRUDENCE region and then assigned to the corresponding grid cells. NAO was estimated at December-to-February (winter), March-to-May (spring) and June-to-August (summer) basis. To determine the strength of the relation between $p(w_i)$ and $y_i$, the regression coefficient $\hat{\beta}_1$ was used, while the statistical significance of the correlation was estimated by the maximum likelihood function on the rejection of the null hypothesis that wetness and the corresponding descriptor $y_i$ are independent.

**Limitations**. Although our results generally agree with recent regional studies (Supplementary Table 2; Supplementary Fig. 10), the above analysis presents some certain limitations that should not be overlooked. The most important factor is the amount of uncertainty involved in palaeoclimatic reconstructions, especially when it comes to gridded data (regression bias). In addition, the OWDA reconstructs summer ("warm season") scPDSI, which is strongly correlated with the annual scPDSI ($\rho = 0.7$; CRU dataset) through prescribed monthly Markovian persistence ($\rho_1 = 0.897$); it is still a source of uncertainty. Finally, the temperature reconstruction is based on a very small subset of four tree-ring chronologies also used in the OWDA of which only three are in the domain of interest here. Even so, this might produce the opposite type of bias, i.e., create spurious correlations between the examined reconstructions. This source of bias is unlikely to be large, however.

**Code availability**. All the analyses have been performed using the statistical software R[72]. The code used in this study, as well as an interactive version of the main figures for further exploration, can be found at https://shiny.fzp.czu.cz/KVHEM/OWDA/.

**Data availability**. The original version of Old World Drought Atlas, as well as the other palaeoclimatic reconstructions (NAO and temperature), used in this study can be found at NOAA/World Data Service for Palaeoclimatology archives. The spatiotemporal subset used in our analyses is publicly available at https://shiny.fzp.czu.cz/KVHEM/OWDA/.

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

## Acknowledgements

This study was supported by the Czech Science Foundation (grant no. 16–16549S) and it consists Lamont-Doherty Earth Observatory contribution no. 8210. The authors would like to thank Dr. C. Pappas for reading the manuscript and providing useful comments and suggestions, as well as Marc-Olivier Goudreault for his feedback on the manuscript's title and abstract.

## Author contributions

Y.M. conceived the idea, performed the analyses and wrote the manuscript. M.H. and P. M. have contributed in the experimental design. All authors have taken part in the discussion of the results and the editing of the manuscript.

## Additional information

**Competing interests:** The authors declare no competing interests.

