## [Peer Review File · Nature Communications]

Reviewers' comments:

Reviewer #1 (Remarks to the Author):

Multidecadal hydroclimatic conditions of Europe in a millennial perspective

Summary

The manuscript presents a study of the Old World Drought Atlas time series, with particular focus of the most recent 90-years. This period is described as being anomalously wet when considered across the entire study area (Europe, with parts of North Africa and the Middle East). The authors note that the "average state" of this index would result in much drier conditions across this area, which would clearly have significant impacts on water resource availability. This is an important point that may be of interest to a wide readership, but the link between this and recent (and projected) climate change is not given much attention. Is the recent 90-year wet period in part a climate change signal, or independent from global climate change? This is an important question. Similarly, reading through the manuscript a number of important and interesting topics and lines of enquiry are raised, but without satisfactorily addressing any of them. For example, breaking the recent 90-year period into three reveals that the spatial pattern of anomalies in the early 20th century are very similar (just weaker) to the current period, which on the surface appears consistent with the pattern of continental (and global) temperature change, but is not commented on in the manuscript. The possibility of a connection to the NAO is suggested, but explored in an overly simplistic manner. Finally the separate roles of thermal vs atmospheric circulation factors on hydroclimatic extremes are considered, but without full analysis.

Specific points

1. P2 line 37. Drier not dryer
2. P2 lines 41. Should be "model-observation integration"?
3. P2 line 55. "help us determine it" – what is "it"?
4. Lines 59-64, Figure 1. When exactly did the dry & wet periods start/end?
5. Lines 72-74. "intense series of succeeding dry decades" [since 1950-1980] – but there have only been two decades since then...
6. Figure 1 caption. Blue crossed squares & orange crossed circles are noted – but the squares are orange and circles are blue. Please correct – at this point it is unclear which symbol is indicative of positive or negative NAO.
7. Lines 81-82. Current pluvial conditions only seem unique in terms of duration of non-exceedance probability. Duration from percentage of cells with positive PDSI as fraction of total number of cells does not seem unique.
8. Lines 84-99, Figures 2 and 3.
 - a. Fig 2. Explain arrow and shading in caption. "grey lines: 992-1922" missing digit, 1892?
 - b. What do the four periods correspond to (the length of the recent wet period?), why 30-year periods?
 - c. Fig 3. 1923-52 and 1983-2012 are very similar spatial patterns, but with higher magnitude dryness and wetness in the latter period... in this context it is the 1953-1982 period that seems anomalous, rather than the current period. This may not be coincidental, but this would approximately match with continental (and indeed global)-scale warming in the early 20th century, stability/cooling in the middle of the century, followed by recent warming. This needs commenting on.
9. Links with the NAO
 - a. Too much made of this relatively short record (in terms of number of events).
 - b. 2 of 3 dry periods are coincident with a negative NAO. 1 of 4 wet periods corresponds with both a positive and negative NAO.
 - c. Whilst I have no problem with the statement that the NAO-related atmospheric circulation anomalies are important for hydroclimate, a far more precise statement (and supporting analysis) needs to follow. The authors state that "half of the extreme hydroclimatic periods occurred during peaks of the NAO", but equally true is that half of the extreme hydroclimatic periods occurred

during neither peaks nor troughs of the NAO, and one further event occurred across both a peak and trough in the NAO index. This is not a robust basis on which a coherent relationship to the NAO can be conjectured.

d. Fig. 4. Parts b and d need better labelling/description, and possibly the colour scheme too. Parts b and d show the confidence intervals (p-values) for the logistic regression? If so, the colour scale on the right of the figure should be labelled as such. The conventional limits for statistical significance need to be better highlighted (such as $p=0.1$, 0.05 and 0.01) – either by isolines, or by changing the continuous colour ramp to three or four clearly distinguishable shades.

e. The NAO is not the only factor that influences atmospheric circulation (and hydroclimate) over Europe (e.g. for droughts: Hannaford et al. 2011, Parry et al. 2012, Kingston et al. 2015), so by focussing only on this pattern only part of the role of atmospheric circulation could ever be revealed.

10. Role of temperature vs atmospheric circulation for hydroclimate. The analysis only covers part of the topic. Firstly, as noted in the previous comment the NAO is only one way in which atmospheric circulation varies. Secondly, temperature is a function of both thermal changes and atmospheric circulation. Temperature with effects of atmospheric circulation variation removed would need to be analysed to get a more complete comparison between role of circulation and temperature for hydroclimatic variation.

References

- Hannaford, J., B. Lloyd-Hughes, C. Keef, S. Parry, and C. Prudhomme, 2011: Examining the large-scale spatial coherence of European drought using regional indicators of precipitation and streamflow deficit. *Hydrol. Processes*, 25, 1146–1162, doi:10.1002/hyp.7725.
- Parry, S., J. Hannaford, B. Lloyd-Hughes, and C. Prudhomme, 2012: Multi-year droughts in Europe: Analysis of development and causes. *Hydrol. Res.*, 43, 689–706, doi:10.2166/nh.2012.024.
- Kingston, D. G., J. H. Stagge, L. M. Tallaksen, and D. M. Hannah, 2015: European-scale drought: Understanding connections between atmospheric circulation and meteorological drought indices. *J. Climate*, 28, 505-516. doi: 10.1175/JCLI-D-14-00001.1

Reviewer #4 (Remarks to the Author):

Review of manuscript NCOMMS-17-08162-T “Multidecadal hydroclimatic conditions of Europe in a millennial perspective” by Markonis et al.

This manuscript presents the European hydroclimatic conditions over the last millennium on multidecadal timescales (10/30yr intervals). The authors found in their study that most regions of Europe experienced unprecedented wetness in the last 90 years (representing the longest lasting pluvial in the last millennium), while only some regions show dry decades (Iberian Peninsula and Mediterranean Sea). This finding is not completely new as this is already shown in the original article (Cook et al. 2015) of the used database (OWDA) but on an annual timescale. However, in this manuscript the authors additionally give information about the spatial extend of wet and dry conditions as they sum up the grid cells which experienced drought/wetness, and perform some additional statistical analyses and compare it with NAO and temperature data. The approach of analyzing the OWDA data and the regional resolution might be novel but the conclusion of the study is not that extraordinary.

For sure, the paper will be of interest to climate scientists (and also other disciplines) and might be influential in the climate change debate. But as the study only looks into 30yr windows, i.e. average conditions, no conclusions about the frequency of drought/extreme events can be drawn but such events can have stronger impacts than average hydroclimatic conditions. Thus, my impression is that the conclusion has to be toned down a little bit.

In general, the paper is well-written, straightforward and organized, and the statistical analysis

seem sound. A positive point of the manuscript is that the authors provide all data and codes used for their study.

Some specific comments:

Line 29: "If not, then we are moving into uncharted waters." This sounds very unprofessional and provocative, reword or remove the sentence. Maybe also reword the sentence before.

Line 73/74: Why is greenhouse gas forcing responsible for more severe dry decades in the eastern Mediterranean? Needs a more detailed explanation.

Line 78: Figure caption: "... The positive (blue crossed CIRCLES) and the negative (orange crossed SQUARES)..." And the grey dashed lines represent standard deviation of what?

Line 85: Make an own sentence out the statement in the brackets.

Line 98: Figure caption: add something like: ...grey lines: different 30 yr intervals for the 992-1922 period... There is another "grey line" interval showing the overall wettest conditions. Please give the period and discuss it. The same for the dry periods highlighted with the blue rectangle. When did they occur? This is not discussed sufficiently in the text.

Lines 135ff and Fig. 4: please give the period for these statistics.

Figures in general: The scale of the y-axis or color-scheme often changes (e.g. Fig. 4, S3, S6) and a uniform scale within one Figure would improve the clarity/readability of the results as the comparison of the data would be easier.

Prague, September 29, 2017

Dear Reviewers,

After careful study of your comments we focused the revision of our manuscript on three major points which helped us to both upgrade the quality of our work and to further extend it into some new results and conclusions. These are:

1. Which is the relationship between 30-year scale and the frequency of drought/extreme events in annual or finer temporal scales in general? (Reviewer #4)
2. Which is the link between the recent 90-year wet period and recent (and projected) climate change? (Reviewer #2)
3. The possible link between hydroclimate and NAO links needs deeper analysis. (Reviewer #2)
 - a. The event-based approach is insufficient due to the short record length.
 - b. Which is the effect of the other circulation patterns/indices?
 - c. What is the role of temperature to drought when the effects of atmospheric circulation variation are removed?

Below, a detailed discussion of these points can be found, as well as our replies to the specific points risen in your reviews. All figures presented here can also be found to the corresponding revised manuscript (RM) and revised supplementary material (RSM). In addition, we have added an interactive representation of the main figures (<https://shiny.fzp.czu.cz/KVHEM/OWDA/>) to communicate our results more effectively and enhance their reproducibility even further.

1. Pluvial/Drought events in finer temporal scales

To investigate the relationship between average (climatic) fluctuations and the frequency of extreme annual drought events the percentage of grid cells with scPDSI in the top/bottom 0.05 and 0.01 quantiles per 30-year period was determined (Figure 1a in the RM). This analysis resulted in two substantial remarks:

- a. During the 1983-2012 the number of grid cells above the 0.95/0.99 quantiles were unprecedented for the whole record, with two other periods (1113 – 1142, 1713 – 1742) experiencing a similar number of annual extremes. [Lines 85-89 in the RM]
- b. Even though the 1953-1982 and 1983-2012 periods are quite similar in the 30-year scale (Figure 1 in the Original Manuscript; OM), there is significant difference in the number of annual extremes. It seems that in the 1923-1952 period, extreme pluvial events were more frequent

even though the climatic conditions were dryer. Similarly, during the 1263–1292 and 1563–1592 pluvials, the annual extremes were closer to their average frequencies. On the contrary, 1113–1142 CE was a period of significantly increased annual maxima, although the 30-year conditions were not that extreme. [Lines 89-96 of the RM]

The last point suggests that an extreme 30-year period is not necessarily a result of subsequent extreme years or vice versa. Therefore, we extended our research and investigated the role of scale in the occurrence of extremes events (Figure 1b in the RM).

To achieve this, we have estimated the mean number of grid points in the 0.95 (0.05) quantile per scale (for the 30-year scale the exact number of grid cells was used). The different behavior of extremes between 30-year scale and the finer ones is quite evident during the last 120 years for pluvials, while dry extremes follow a more uniform pattern (Figure 3bc in the RM). If the whole record is examined, though, periods that extremes propagate in different time scales can be found, suggesting either multi-scale fluctuations either periods of hydroclimatic polarization. The latter could occur either in the time or the space domain, i.e., simultaneous droughts and pluvials in different regions of Europe or shifting between dry and wet periods in finer temporal scales (e.g. 5 years of annual maxima, followed by 5 years of minima resulting in an aggregated 10-year period close to the mean). [Lines 97-104 of the RSM]

Figure 1 [1 in RM]. Number of grid cells of annual scPDSI above (below) 0.95 and 0.99 (0.05, 0.01) quantiles per 30-year period.

Figure 2 [2 in RM]. Percentage of grid cells **b.** above (**c.** below) the 0.95 (0.05) quantile (dashed line) at 1-, 2-, 3-, 5-, 10-, 15-, 30-year scales per each 30-year period.

2. Recent wet period and global warming

The scale analysis of the extremes also allowed us to shed more light into the relationship between the recent wet period and the increase in global (and continental) temperature. Indeed, as Reviewer #2 underlines, *“the spatial pattern of anomalies in the early 20th century is very similar (just weaker) to the current period, which on the surface appears consistent with the pattern of continental (and global) temperature change”* [Lines 125-127 in the RM]. Combining this remark with the scale analysis of extremes that was suggested by Reviewer #4, one can notice that the signal in variation is similar in all scales except the 30-year for pluvial conditions (Figure 2b in the RM). This suggests that during this period there are, at least, two main drivers of hydroclimatic variability; one following directly the temperature dynamics affecting finer scales, and one manifesting in the longer ones. This slower component seems to become progressively dominant and in 1983-2012 CE the cell percentage increases with scale for both pluvial and dry conditions. Similar findings have been reported in the study of drought in regional model simulations, where the future changes in the longer scales are found to be more pronounced than the shorter ones [Heinrich and Gobiet 2012]. Another likewise example can be found in the well-studied California drought; the warming component of the drying process appears to be slowed down by natural variability, but still evident as a background trend [Williams et al. 2015]. [Lines 128-138 in the RM]

On the other hand, during the 1923-1952 interval, the 30-year component seems to be inactive. Further investigation (Figure S7 in the RM) demonstrated that this is due to the succession of years with very high and very low number of grid cells with extreme values. This temporal polarization in the annual scale resulted to the complete lack of extremes in the 30-year scale. Since 1973, the years where the annual extremes are minimal are not close to zero, indicating that during the since 1970s years even in the driest years there are always some regions that experience extreme annual wet conditions. This small shift in the minima was found to be unprecedented in the whole record in terms of duration, i.e. 40 years with at least twenty grid cells per year at the highest 0.95 quantile of wet conditions. [Lines 149-158 in the RM]

Additionally, the comparison of the recent hydroclimatic temporal dynamics with the other pluvials, advocates to the fact that the recent pluvial has the most persistent progression and only during 1113-1142 CE and 1713-1742 CE the same pattern in the frequency of extremes across scale was observed (Figure S8 in the RM). According to the Luterbacher et al. reconstruction, temperature has also risen in these two periods (1117-1157 CE: 1.2 degrees and 1697-1727 CE: 0.8 degrees), presenting some link

between them. However, closer examination revealed other periods with similar thermal changes without a corresponding increase in the pluvial extremes (e.g. 1217-1257 CE or 1127-1157 CE). [Lines 159-167 in the RM]

In any case, the above results highlight the need of analysing the influence of temperature to droughts/pluvials, especially regarding the future climatic change. Since the validity of current model simulations for precipitation is still significantly debated [Trenberth et al. 2017], we should try to understand to what extent the projected increase in temperature, which is currently better represented by Earth system models, might affect the forthcoming hydroclimatic regime. [Lines 168-173 of the RM]

Figure 3 [S7 in the RSM]. Number of grid cells above the 0.95 quantile (dashed line) at 1-, 10-, and 30-year scales.

Figure 4 [S8 in the RSM]. As Figure 3bc but different axes are used to examine the scale effect in order to explore the temporal progression of the older pluvials.

3. Links with NAO

Substantial concerns have been risen by Reviewer #2 about our approach on the exploration of the NAO effect on hydroclimatic conditions; especially for lines 122-131 in the OM. We agree that this can be regarded as simplistic analysis, but it is used as an introduction to the logistic regression approach and therefore we have reported our initial findings as *“indicative of possible interference”* [Line 127 of OM] resulting to *“no distinct signal could be detected”* for NAO and temperature [Lines 130-131]. However, we understand that this formulation might be confusing and seen as naive. Therefore, we have decided to remove these lines and the Trouet et al. dataset and focus only in the Luterbacher et al. record, enhancing our investigation with Empirical Orthogonal Function (EOF) decomposition and wavelet analysis.

We have also investigated the other known factors that influence atmospheric circulation (and hydroclimate) over Europe include Arctic Oscillation (AO), the Scandinavian pattern (SCA) and Eastern Atlantic/Western Russia pattern (EA/WR) [Hannaford et al. 2011, Parry et al. 2012, Kingston et al. 2015, Steirou et al., 2017, and references therein]. However, since there is no paleoclimatic reconstruction of these indices it is difficult to robustly determine their influence to European hydroclimate through OWDA. To overcome this setback, one could compare the EOF patterns to the corresponding spatial correlation patterns of the observational records of the indices, as suggested by Baek et al. [2017]. We must note though that this is representative only for the periods that the global circulation retains the features of the last 50 years, i.e. the observational record length, and not for periods when shifts in teleconnection patterns are observed [e.g. Fischer, 2001; Rimbu et al., 2003]. Encouragingly, recent findings [Edwards et al., 2017] suggest that at least for Medieval Climate Anomaly and Little Ice Age, modern analogues of atmospheric circulation patterns can be found for Europe. [Lines 218-230 in the RM]

The first nine EOFs of the scPDSI have been determined, explaining approximately half of the observed variance (54%) and depicting some distinct spatial patterns (Figure 5 in the RM). It is worthy to note the striking analogy between EOF1/EOF2 and Figures 4cd, corresponding to the NAO correlation (33% of variance), while the EOF4 pattern can be found within the temperature regions of influence (Figures 4ab in the RM). On the contrary, the EOF modes above 6 represent rather local spatial patterns (6% of variance). The comparison between the AO and SCA correlation patterns with the CRU scPDSI dataset for the last 50 years, suggests that they are similarities with the EOFs 4, 5 and 6 of OWDA's scPDSI (8% of variance; Figure S12 in the RSM), confirming the previous studies [Hannaford et al. 2011, Parry et al.

2012, Kingston et al. 2015]. Similarly to the case of NAO, the distinction between the circulation indices and temperature is quite obscure. For example, SCA (Figure S11a,c,e) seems to present the same pattern with temperature (Figures 4ab in the RM) and EOF4. On the other hand, most of the indices are correlated with temperature only for central to northern Europe (NAO is similar to AO; not shown here), suggesting different physical processes in hydroclimatic conditions over southern Europe. Correspondingly to the NAO, the rest of the major climatic indices confirm the strong interaction between temperature, precipitation and atmospheric circulation. [Lines 237-253]

These findings highlight the importance of investigating the effect of other circulation patterns and further discriminating the role of temperature without the effect of NAO, as has already been suggested by Reviewer #2.

A formidable way to distinguish the influence of atmospheric circulation and the direct thermic effect of temperature is to investigate the time scales that each of these components appears to be more active and compare them with the corresponding time scales of the drought fluctuations. This can be achieved by wavelet analysis in the most significant EOFs and straightforward comparison with the wavelets of temperature and NAO (Figure 6 in the RM). It is easy to notice that temperature presents more intense fluctuations at lower frequencies than NAO, which fluctuates mostly in the 4- to 32-year domain, which is also in good terms with the logistic regression outcome. However, most of the EOF modes reveal a similar structure to temperature (especially modes 2 and 4), while EOF 1 resembles more to NAO. Still, the reconstructed NAO record has shorter length compared to temperature and lower frequencies might be underrepresented in wavelet analysis. Other palaeoclimatic reconstructions [Trouet et al. 2009, Ortega et al. 2016], show fluctuations in longer time scales, and thus their wavelets resemble more with the temperature (they do not sufficiently describe high frequencies, though, since they are smoothed). [Lines 254 -268 in the RM]

Figure 5 [5 in the RM]. Spearman's linear correlation coefficients between scPDSI and the base point of the 9 first modes of the empirical orthogonal transformation [van den Dool et al., 2000].

Figure 6 [S12 in the RSM]. Correlation coefficients (0.9 significance) between CRU scPDSI 3.25 gridded dataset and annual **a.** SCA and **b.** AO. Similarly for precipitation (**c.**, **d.**) and temperature (**e.**, **f.**) (EOBS 15.0 dataset). Data extraction and analysis were performed in the KNMI explorer on-line platform.

At this point, the influence of the Atlantic oceanic circulation, as presented by Atlantic Multidecadal Oscillation (AMO) index should also be considered, as it is known to influence summer European hydroclimate [Sutton and Hodson 2005] or continental drought events in general [Sheffield et al. 2009], and also exhibit strong variability in the ~55- to 70-year time domain [Knudsen et al. 2011]. The latter matches well with all EOF modes suggesting that the multi-decadal modulator of European drought might lie in the oceanic circulation, similarly to the long-term correlation of drought in the United States [McCabe et al. 2004], which has also been increasing during the latter half of the 20th century [Andreadis and Lettenmaier 2006]. In addition, it can be easily seen (Figure S13 in the RM), that during both 1923-1952 and 1983-2012 periods, the AMO was shifting from negative to positive phase, whereas the opposite transition dominated the intermediate interval. [Lines 269 - 278 in the RM]

Figure 7 [6 in the RM]. Wavelet analysis of winter NAO, temperature and six first EOF modes of scPDSI. The thick black contour designates statistical significance ($p > 0.95$) and the cone of influence, where edge effects can potentially distort the picture, is shown as a lighter shade.

Concluding, the revised version of the manuscript based on the extra analyses suggested in the reviews confirmed the initial findings and revealed some novel substantial points. These are the scale

dependency in the propagation of extremes [Lines 283-295] and the coupling between temperature and atmospheric/oceanic circulation that remains indistinguishable [Lines 296-310]. We hope that you will find our effort within the spirit of the suggested revision and consider our work suitable for publication.

Sincerely,

On behalf of all co-authors,

Yannis Markonis

References

All references can be found in the RM.

Addressing specific points

Reviewer #2

1. P2 line 37: Drier not dryer

It has been corrected.

2. P2 lines 41: Should be “model-observation integration”?

It has been corrected.

3. P2 line 55: “help us determine it” – what is “it”?

It refers to “*significance of the observed change*”. It has been reformulated to “*However, the limited length of observational records hinders our ability to assess the significance of the observed change in the long-term context. The OWDA, or similar reconstructions, could help us determine its significance, by comparing the recent variability to the multi-decadal fluctuations of the past*”

4. Lines 59-64, Figure 1: When exactly did the dry & wet periods start/end?

Figure 1 was revised and the original has been transferred to the RSM. Periods in manuscript are now referred in the RM with their starting and ending year, e.g., Lines 89-94 of the RM.

5. Lines 72-74: “intense series of succeeding dry decades” [since 1950-1980] – but there have only been two decades since then...

Here we are referring to the three succeeding decades 1980-1990, 1990-2000 and 2000-2010 and therefore we have reformulated our phrase to: “*it was succeeded by three rather dry decades*” (Lines 80-81 in the RM).

6. Figure 1 caption: Blue crossed squares & orange crossed circles are noted – but the squares are orange and circles are blue. Please correct – at this point it is unclear which symbol is indicative of positive or negative NAO.

The whole part with the Trouet et al. reconstructions has been removed as discussed above.

7. Lines 81-82: Current pluvial conditions only seem unique in terms of duration of non-exceedance probability. Duration from percentage of cells with positive PDSI as fraction of total number of cells does not seem unique.

The phrase was revised accordingly: “*For most of Europe, current pluvial conditions seem to be unique in terms of duration of non-exceedance probability for the last thousand years, when observed in 30-year time scale (Figure 1a).*” (Lines 86-88 in the RM).

8. Lines 84-99, Figures 2 and 3:

- a. Fig 2. Explain arrow and shading in caption. “grey lines: 992-1922” missing digit, 1892?
- b. What do the four periods correspond to (the length of the recent wet period?), why 30-year periods?
- c. Fig 3. 1923-52 and 1983-2012 are very similar spatial patterns, but with higher magnitude dryness and wetness in the latter period... in this context it is the 1953-1982 period that seems anomalous, rather than the current period. This may not be coincidental, but this would approximately match with continental (and indeed global)-scale warming in the early 20th century, stability/cooling in the middle of the century, followed by recent warming. This needs commenting on.

a. To explain arrow and shading we have added: *“The blue arrow represents the temporal evolution of last three 30-year intervals. The dry periods are also highlighted by the blue transparent rectangle.”* The “grey lines: 992-1922” has been rephrased as *“different 30-year intervals for the 992–1922 period”*, as also suggested by Reviewer #4).

b. 30-year periods correspond to climatic conditions. To clarify it we have rephrased: *“For most of Europe, current pluvial conditions seem to be unique in terms of duration of non-exceedance probability for the last thousand years, when observed in 30-year (climatic) time scale (Figure 1a).”* [Lines 86-88 of the RM]

c. This is a very interesting remark that has been further analyzed and discussed in lines 123-158 of the RM (see also the discussion above).

9. Links with the NAO:

- a. Too much made of this relatively short record (in terms of number of events).
- b. 2 of 3 dry periods are coincident with a negative NAO. 1 of 4 wet periods corresponds with both a positive and negative NAO.
- c. Whilst I have no problem with the statement that the NAO-related atmospheric circulation anomalies are important for hydroclimate, a far more precise statement (and supporting analysis) needs to follow. The authors state that “half of the extreme hydroclimatic periods occurred during peaks of the NAO”, but equally true is that half of the extreme hydroclimatic periods occurred during neither peaks nor troughs of the NAO, and one further event occurred across both a peak and trough in the NAO index. This is not a robust basis on which a coherent relationship to the NAO can be conjectured.
- d. Fig. 4. Parts b and d need better labelling/description, and possibly the colour scheme too. Parts b and d show the confidence intervals (p-values) for the logistic regression? If so, the colour scale on the right of the figure should be labelled as such. The conventional limits for statistical significance need to be better highlighted (such as $p=0.1$, 0.05 and 0.01) – either by isolines, or by changing the continuous colour ramp to three or four clearly distinguishable shades.
- e. The NAO is not the only factor that influences atmospheric circulation (and hydroclimate) over Europe (e.g. for droughts: Hannaford et al. 2011, Parry et al. 2012, Kingston et al. 2015), so by focusing only on this pattern only part of the role of atmospheric circulation could ever be revealed.

a., b., c., e. : Following the reviewer’s points the whole Section has been reconstructed (see discussion above).

d.: The Fig. 4. Parts b and d of the OM (Figure S9 in the SRM) were revised accordingly.

10. Role of temperature vs atmospheric circulation for hydroclimate: The analysis only covers part of the topic. Firstly, as noted in the previous comment the NAO is only one way in which atmospheric

circulation varies. Secondly, temperature is a function of both thermal changes and atmospheric circulation. Temperature with effects of atmospheric circulation variation removed would need to be analysed to get a more complete comparison between role of circulation and temperature for hydroclimatic variation.

As explained in the previous comment, the whole Section has been reconstructed (see discussion above).

Reviewer #4

Line 29: “If not, then we are moving into uncharted waters.” This sounds very unprofessional and provocative, reword or remove the sentence. Maybe also reword the sentence before.

The sentence has been revised to “*The opposite would mean that current conditions could persist for the next decades or deviate even further.*”

Line 73/74: Why is greenhouse gas forcing responsible for more severe dry decades in the eastern Mediterranean? Needs a more detailed explanation.

We have reformulated the phrase to “*possibly due to increased evapotranspiration linked to global warming*”. [Lines 82-83 in the RM].

Line 78: Figure caption: “... The positive (blue crossed CIRCLES) and the negative (orange crossed SQUARES) ...” And the grey dashed lines represent standard deviation of what?

The circles and squares were removed as the whole part with the Trouet et al. reconstructions has been removed as discussed above. The figure was moved in the RSM, where “*of the scPDSI for Europe at 30-year scale.*” was added in the caption.

Line 85: Make an own sentence out the statement in the brackets.

The statement has been reformulated out of the brackets [Lines 110-111 in the RM].

Line 98: Figure caption: add something like: ...grey lines: different 30 yr intervals for the 992-1922 period... There is another “grey line” interval showing the overall wettest conditions. Please give the period and discuss it. The same for the dry periods highlighted with the blue rectangle. When did they occur? This is not discussed sufficiently in the text.

We have added the phrase “*different 30-year intervals for the 992–1922 period*”. The other “grey line” refers to period 1713 – 1742, and it has been discussed in the RM as: “*The only other 30-year interval which clearly surpasses them is the period between 1713 and 1742, with a median grid cell scPDSI quantile equal to 0.81; significantly higher than the 0.76 corresponding to the 1983-2012 interval.*” [Lines 116-119 in the RM]. The dry periods are properly referenced in the RM as “*These 30-year dry intervals can be found within the 1023–1082, 1413–1472 and 1803–1862 periods (Figure S3a)*” [Lines 121-122 in the RM].

Lines 135 and Fig. 4: please give the period for these statistics.

The period (1500 and 2012) was added in the RM. [Line 234 in RM]

Figures in general: The scale of the y-axis or color-scheme often changes (e.g. Fig. 4, S3, S6) and a

uniform scale within one Figure would improve the clarity/readability of the results as the comparison of the data would be easier.

After the addition of the new analyses, we have tried to focus on the grid cells percentage metric to the point that this was possible. The scale of y-axis Figure S3 in the OSM [S3b in the RSM] has been corrected to match Figure S3a in RSM. We have tried to use the blue/brown color theme to scale pluvial/wet conditions and the red/blue for the other scales.

Reviewers' comments:

Reviewer #1 (Remarks to the Author):

Review of revised version of "Multidecadal hydroclimatic conditions of Europe in a millennial perspective"

This revised manuscript is much improved from the initial submission. I find the responses to my previous comments and suggestions satisfactory. Furthermore, the substantial changes and extensions to the manuscript and analyses reported therein have made the key components of the manuscript more convincing, and likely of wider interest too. I have no major suggestions for further changes, and just two comments that the authors may wish to consider:

1. The wavelet analysis is a useful addition, but the authors may be able to make a more convincing case for the presence of relationships between the NAO, temperature and the EOFs if cross-wavelet analysis was used to directly compare timing and periodicity of variation in these time series, rather than the more qualitative visual comparison performed in the manuscript at present.

2. The consideration of the AMO in the text and supplementary figure 13 is interesting and warrants further discussion – especially given that one of the main findings from this work seems to be that there is some uncertainty as to the causes of the recent departure from average hydroclimatic conditions.

Reviewer #5 (Remarks to the Author):

This is a very interesting re analysis of the OWDA and adds some important knowledge about Europe's hydroclimate past. The authors have partially addressed the original reviewers comments, although I feel that some of the original points, particularly of reviewer 2 are somewhat overlooked. Whilst the additional analysis is very detailed the results of it unfortunately, in my view, weaken the overall findings and require a longer format to explore fully. This is a really nice paper but its more suited to a long format climate journal i feel.

As this is a resubmission I have reviewed with an eye to the original reviewers comments.

Reviewer 1 – point out an issue with the author’s identification that the last 90 years has been wet across Europe in comparison to the length of the OWDA and that the author’s state that, should this change to the average OWDA conditions going forwards, there would be significant implications for water resources in Europe. Obviously this is a major research point of the paper and the original reviewer 1 did not feel that this was dealt with in the manuscript.

I feel reviewer 1’s points below have not been addressed

- 1.what is the broad link between the recent wetter 90-year period and relevant climate projections?
- 2.is the recent 90-year wet period in part a climate change signal, or independent from global climate change?
- 3.the spatial pattern of hydroclimate anomalies in the early 20th century are similar to the current period, which on the surface appears consistent with the pattern of continental (and global) temperature change, this needs commenting on?
4. The connection to the NAO needs a more nuanced exploration.
5. thermal vs atmospheric circulation impacts on hydroclimatic extremes are considered, but without full analysis.
6. Reviewer 4 feels that, due to the segment lengths are fairly limited the conclusions need to be toned down. The reviewer does not state whether they think that such a toning down would make the article no longer suitable for the target journal but they do state that the findings are primarily of interest to the climate dynamics communities.

I feel points 1,5 and 6 have not been addressed fully in the resubmitted manuscript (although there is some additional analysis relevant to point 1 above). I feel that points 2,3,4 have been addressed with some considerable analysis, however, this has somewhat decreased the potency of the arguments in the paper.

In light of the above I do not feel that the paper is suitable for this journal and that it would be better suited to a long format climate journal.

More detailed comments.

The opening paragraph is rather clumsy. I would suggest working with an editing service to refine the writing. There are some suggestions below too.

Opening paragraph.

~~During the last several years, there has been g~~ There is growing concern about the effect 11 of global warming on water resources, especially at continental and regional 12 scales. ~~The last-The~~ The IPCC report ~~on extremes suggests that there is only~~ medium 13 confidence about an increase in droughts over Europe during the last century however the palaeoclimate perspective on hydrological extremes might add nuance to our understanding of drought risk. 14 Here we show that in the last 90 years, dryness has decreased significantly 15 over much of Europe, resulting in a wet period of unprecedented length over 16 the last millennium. Using the tree-ring reconstruction derived Old World Drought Atlas, the magnitude of past 17 long-term hydroclimatic fluctuations ~~was-is explored investigated~~ and other similar hydroclimatic 18 intervals in terms of severity and geographical extent ~~were-are~~ identified. None of 19 them, however, matched the current pluvial period in duration. Plausible explanations 20 for this persistent change, such as the intensification of the global 21 hydrological cycle due to temperature increase and the role of 22 atmospheric/oceanic circulation were explored (this is too vague). Even though these factors 23 appear to be correlated to the

regional hydroclimatic conditions, these 24 relationships are found to be weak, capturing only a fraction of the observed 25 variability. The recent persistent departure may strengthen uncertainty on how 26 Europe's hydroclimate will fluctuate during the following decades. If European hydroclimate returns 27 to its average state from the last millennium, then much dryer conditions should be expected in central 28 and northern Europe and much wetter to the South. The opposite would mean 29 that current conditions could persist for the next decades or deviate even 30 further.

Reviewer 1

This revised manuscript is much improved from the initial submission. I find the responses to my previous comments and suggestions satisfactory. Furthermore, the substantial changes and extensions to the manuscript and analyses reported therein have made the key components of the manuscript more convincing, and likely of wider interest too.

We would like to thank the Reviewer for her/his constructive comments that helped us to make all these changes and make the key components of the manuscript more convincing, and likely of wider interest too.

I have no major suggestions for further changes, and just two comments that the authors may wish to consider:

1. The wavelet analysis is a useful addition, but the authors may be able to make a more convincing case for the presence of relationships between the NAO, temperature and the EOFs if cross-wavelet analysis was used to directly compare timing and periodicity of variation in these time series, rather than the more qualitative visual comparison performed in the manuscript at present.

We have replaced the wavelet with cross-wavelet analysis (Lines 364-382 and Figures 9 and 10 in the revised manuscript) which allowed us for a more straightforward comparison between NAO, temperature and EOFs, as well as AMO (see reply to comment #2).

2. The consideration of the AMO in the text and supplementary figure 13 is interesting and warrants further discussion – especially given that one of the main findings from this work seems to be that there is some uncertainty as to the causes of the recent departure from average hydroclimatic conditions.

We have reformatted and moved Fig. S13 of the original manuscript from Sup. Material to main text and also included AMO in the cross-wavelet analysis. In addition we have extended the discussion about the role of AMO in Europe's hydroclimate and its strong coupling with temperature (Lines 353-382 in the revised manuscript).

Reviewer 5

Reviewer 1 – point out an issue with the author's identification that the last 90 years has been wet across Europe in comparison to the length of the OWDA and that the author's state that, should this change to the average OWDA conditions going forwards, there would be significant implications for water resources in Europe. Obviously this is a major research point of the paper and the original reviewer 1 did not feel that this was dealt with in the manuscript.

It is unfortunate that the Reviewer #2 could not evaluate our replies to her/his comments and we hope that in this revision we further elaborate on the concerns raised by Reviewer #5 (mainly for points 1, 5, 6). In term of new analyses, the cross-wavelet methodology has been used to further pinpoint the scales of co-variability between temperature, NAO and AMO (also added during this revision) and the leading EOFs of the scPDSI. Another important addition was the recent findings of Kröner et al. (Reference 48 in the revised manuscript), who decomposed the thermal (TD) and circulation (CO) components of hydroclimatic change, presented in the figure below.

It can be seen that at high latitudes, the dominant wet conditions are driven by atmospheric circulation, amplified by the thermal effect in terms of both increased precipitation and zonal humidity transport. Over France and Germany there is a mixed impact of temperature and circulation, reflected also by conflicting results in relevant studies (References 48, 51 in the revised manuscript). Finally, in southern Europe, the two components oppose each other, but the thermal effect does not compensate for the circulation effect.

In addition, we highlight two other recent studies about the future evolution of atmospheric circulation model projections on atmospheric circulation (References 51, 52 in the revised manuscript). In the Figures below, it can be seen that the north-south pattern is reversed in individual models or ensemble members (second figure presents precipitation change %). Finally, we have also enriched our discussion in the north-to-south pattern, confirming the results from some other recent studies (References 42, 43 in the revised manuscript).

I feel reviewer 1's points below have not been addressed

1. what is the broad link between the recent wetter 90-year period and relevant climate projections?

Although it does not lie within the prime scope of this paper to discuss future climate projections, we understand that the statement in the first paragraph (Lines 26-29 in the original manuscript) about future changes to the European hydroclimate could be supported more in the manuscript. We have found our results to be in agreement with drought projections of regional climate models for Europe (References 24, 44 in the revised manuscript). However, model projections on atmospheric circulation are quite ambiguous and we are still far from any confident statements about the future change of Europe's hydroclimate. As aforementioned, in two independent studies (References 51, 52 in the revised manuscript), it has been demonstrated that the north-south pattern of persisting wetness and dryness is reversed in individual ensemble members or models, depending on the long-term evolution of atmospheric dynamics (Lines 288-324 of the revised manuscript).

2. is the recent 90-year wet period in part a climate change signal, or independent from global climate change?

We have shown that this 90-year wet period is linked by both changes in temperature and circulation (Lines 288-340 of the revised manuscript) and the relationships may differ across scales (Lines 341-353 of the revised manuscript). Thus it is not plausible to make strong statements about its link with global warming. However, we discuss the resemblance between the pattern of global temperature increase and the spatial pattern of hydroclimatic change (as addressed in the comment #3).

3. the spatial pattern of hydroclimate anomalies in the early 20th century are similar to the current period, which on the surface appears consistent with the pattern of continental (and global) temperature change, this needs commenting on?

We underline the strong resemblance between the global temperature change and the spatial pattern of hydroclimatic change. Thus, we have reformulated Lines 119-128 in the revised manuscript:

Another noteworthy result stemming from the scale analysis of extremes, considers the relationship between the recent wet period and the increase in global (and continental) temperature. The spatial pattern of anomalies (Figure 3) in the early 20th century is very similar to the current period and follows the increase of global temperature change, i.e., increase during 1923-1952 and 1983-2012 and slight decline during 1953-1982. Notably, the areal extent of pluvial conditions fluctuates similarly in all but the 30-year time scale (Figure 2b). This suggests that during this period there are, at least, two main drivers of hydroclimatic variability; one following directly the temperature dynamics affecting finer scales, and one manifesting in the longer ones.

To further clarify this, we have added to the concluding section of our manuscript the following phrase (Lines 394-397 in the revised manuscript):

The extremity of the current multi-decadal conditions involves a discrete north-to-south dipole, which follows both the 20th century global temperature increase and the long-term atmospheric/oceanic circulation variability

4. The connection to the NAO needs a more nuanced exploration.

We have added the cross-wavelet technique to explore the relationships between the NAO, temperature and the EOFs. Our findings suggest that there is some evidence that atmospheric coupled with oceanic circulation could be the main driver of droughts/pluvials over Europe.

We have also applied the entropy mutual information and convergent cross-mapping methods, which did not provide any additional findings and therefore they were omitted.

5. thermal vs atmospheric circulation impacts on hydroclimatic extremes are considered, but without full analysis.

We have added the cross-wavelet technique and included the AMO to explore also the role of oceanic circulation as suggested by Reviewer #1. Most importantly, we have found a recent decomposition of the thermal vs atmospheric circulation components and argued how it is linked with our results (Lines 325-339 in the revised manuscript):

Our results confirm the strong coupling between the thermal and the dynamic effect. Since the OWDA is a reconstruction of self-calibrated Palmer drought index derived from tree-rings, inevitably the influence of soil moisture availability is also encompassed in the hydroclimatic signal of the proxy records. Therefore, it is hard to distinguish the explicit role of temperature versus atmospheric circulation and establish links of causality between them. In light of the findings of Kröner et al. (Reference 48 in the revised manuscript) on the decomposition of the thermal and circulation effect, though, we can highlight three different regimes. At high latitudes, the dominant wet conditions are driven by atmospheric circulation, amplified by the thermal effect in terms of both increased precipitation and zonal humidity transport. Here, these two components act synergistically (see also Figure 6 in the revised manuscript). Over France and Germany there is a mixed impact of temperature and circulation, reflected also by conflicting results in relevant studies (References 48, 51 in the revised manuscript). In southern Europe, the two components oppose each other, but the thermal effect does not compensate for the circulation effect. Additionally, there is a positive feedback of soil moisture to precipitation by reducing the regional recycling of water, which further amplifies the dry conditions (Reference 53 in the revised manuscript).

6. Reviewer 4 feels that, due to the segment lengths are fairly limited the conclusions need to be toned down. The reviewer does not state whether they think that such a toning down would make the article no longer suitable for the target journal but they do state that the findings are primarily of interest to the climate dynamics communities.

Indeed, this is why we have also included the 1-, 2-, 3-, 5-, 10-, 15-, 30-year scales in our revised manuscript. We believe that the decision of the Reviewer #4 to recommend our work for publication in Nature Communications, now states clearly what she/he thinks about the toning down.

I feel points 1, 5 and 6 have not been addressed fully in the resubmitted manuscript (although there is some additional analysis relevant to point 1 above). I feel that points 2, 3, 4 have been addressed with some considerable analysis, however, this has somewhat decreased the potency of the arguments in the paper.

We hope that the revision of our manuscript fully addressed points 1, 5 and 6 and clarified any misconceptions about points 2, 3 and 4.

In light of the above I do not feel that the paper is suitable for this journal and that it would be better suited to a long format climate journal.

Following the Editor's clarification on the contents allowance of Nature Communications, we have extended the original manuscript to more than 4000 words in length, transferred the methods session to the main text from the supplementary material (adding another 1000 words) and increased the number of figures to 10 and references to 72. Since we had a positive review from Reviewer #1, we avoided changing the main structure of the manuscript and focused in elaborating more about our results. Hence, the new content can be mainly found in the discussion (and the methods) section.

More detailed comments.

The opening paragraph is rather clumsy. I would suggest working with an editing service to refine the writing. There are some suggestions below too.

Opening paragraph.

We do not think that our opening paragraph is clumsy. We appreciate the effort of Reviewer #5 to improve it and understand hers/his concerns, since the opening paragraph should feature the paper's content in a concise and comprehensive manner. However, we cannot accept all of hers/his recommendations, because some of them do not help us upgrade our text. Finally, we have read thoroughly the manuscript and improved its syntax and corrected a few typos that were missed in the previous versions of the manuscript.

REVIEWERS' COMMENTS:

Reviewer #5 (Remarks to the Author):

Review.

Nature comms review.

March 2018

The response to reviewers comments is thorough and, whilst the authors, have a tendency to throw copious amounts of new analyses at a constructive point from the reviewers, rather than address the wider point of debate raised, they have clearly put a huge amount of effort in to the construction of a revised manuscript. I maintain that the manuscript, both in terms of subject area, which is really of relevance to the climate dynamics community, and format, is better suited to a long format journal. However, I leave that decision to the editor.

Opening paragraph.

During the last several years, there has been growing concern about the effect of 7 global warming on water resources, especially at regional and continental scales. 8 The last IPCC report on extremes suggests that there is medium confidence about 9 an increase in droughts over Europe during the last century. Here we show that in 10 the last 90 years, dryness has decreased significantly over most of central and 11 northern Europe, resulting in a wet period of unprecedented length over the last 12 thousand years.

I would recommend the below changes for clarity and language.

In recent years, there has been growing concern about the effect of 7 global warming on water resources, especially at regional and continental scales. 8 The most recent IPCC report on extreme events states that there is medium confidence about 9 an increase in European drought frequency during the last century. Here we show that in 10 the last 90 years, dryness has decreased significantly over most of central and 11 northern Europe, resulting (the word resulting here infers that the dry period caused the wet period, better to use 'reveals' or an equivalent) in a wet period of unprecedented length over the last 12 thousand years.

There is a real need for clarity in the writing in this one paragraph, it's just not up to NPG standards for clear expression. I am sure the editor can work with the authors to rectify that.

Line 29-30, expression here is also somewhat awkward. Rewrite for clarity of meaning.

Line 38- expand what is meant by probabilistic inference to indicate a discussion of causal mechanisms and links to large scale atmospheric features of interest.

Line 46, the current hydro climatic regime and specifically placing it in its long term context.

Line 66- some persisting wet behaviour. Again this is grammatically poor and lacks clarity. Rewrite for clarity of meaning and expression.

Line 133, formatting error, extra line.

The discussions related to dynamics are much improved and of genuine insight.

Discussions and explorations around the NAO are also much improved.

There are still limitations with the investigation that relate to the OWDA itself, the complexity of the dynamics being explored and some of the steps towards conclusive statements made. I feel the paper is better suited to a climate journal. That said the level of analysis in the paper, and behind it, is thorough and robust and the authors have made a genuine attempt to address a complex suite of reviewers comments. If the editor decides to publish the manuscript I would, however, recommend the authors are encouraged to increase the clarity, grammatical consistency and flow of the writing.

There is a real need for clarity in the writing in this one paragraph, it's just not up to NPG standards for clear expression. I am sure the editor can work with the authors to rectify that.

The first paragraph has been rewritten and its size reduced to comply with NCOMM standards.

Line 29-30, expression here is also somewhat awkward. Rewrite for clarity of meaning.

The text was rephrased to:

Understanding the spatial and temporal patterns of the hydroclimatic variability has been one of the most challenging subjects in contemporary climatic and hydrological research, due to its numerous components and the non-linear processes involved.

Line 38- expand what is meant by probabilistic inference to indicate a discussion of causal mechanisms and links to large scale atmospheric features of interest.

The text was rephrased to:

Palaeoclimatic reconstructions could potentially serve as the required basis not only for model-observation integration, but also for probabilistic inference, e.g., for estimation of return period of extreme events.

Line 46, the current hydro climatic regime and specifically placing it in its long term context.

The suggested phrase has been added to the text.

Line 66- some persisting wet behaviour. Again this is grammatically poor and lacks clarity.

Rewrite for clarity of meaning and expression.

The text was rephrased to:

The only wet interval with similar duration of the current pluvial, can be found in 1113-1172, but the 30-year deviations were not so severe and spatially spread.

Line 133, formatting error, extra line.

The extra lines were removed.

If the editor decides to publish the manuscript I would, however, recommend the authors are encouraged to increase the clarity, grammatical consistency and flow of the writing.

The text has been reshaped accordingly.